# Open-TeleVision: Teleoperation with Immersive Active Visual Feedback

**Xuxin Cheng**[*1]   **Jialong Li**[*1]   **Shiqi Yang**[1]   **Ge Yang**[2]   **Xiaolong Wang**[1]

UC San Diego[1]   MIT[2]

https://robot-tv.github.io/

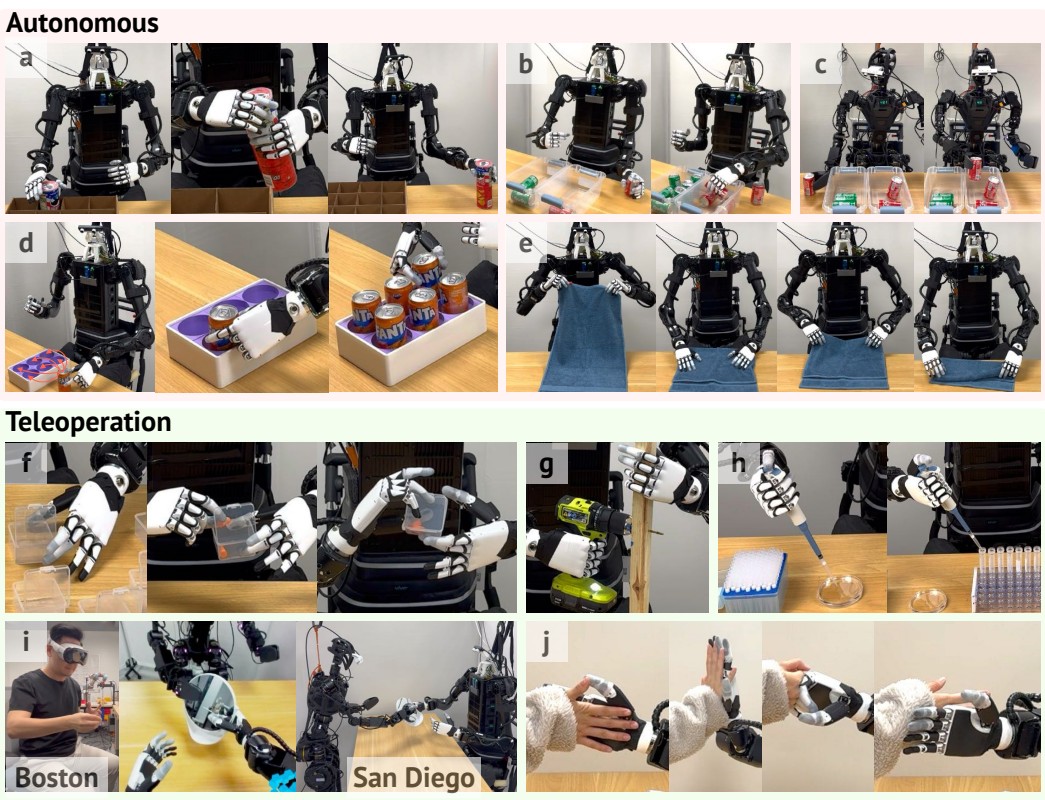

Figure 1: **Autonomous and teleoperated sessions using our setup.** *a-e*: robots executing long-horizon precision-sensitive tasks autonomously. *f-j*: robots executing fine-grained tasks with our immersive teleoperation system. *a*: unloading, in-hand passing; *b*: H1 can-sorting; *c*: GR-1 can-sorting; *d*: can-insertion; *e*: towel folding; *f*; earplugs packing; *g*: drilling; *h*:pipetting; *i*: two operators teleoperate two robots interactively. The operator of H1 robot is approximately 3000 miles away from the GR-1 operator. *j*: interactions with humans.

**Abstract:** Teleoperation serves as a powerful method for collecting on-robot data essential for robot learning from demonstrations. The intuitiveness and ease of use of the teleoperation system are crucial for ensuring high-quality, diverse, and scalable data. To achieve this, we propose an immersive teleoperation system **Open-TeleVision** that allows operators to actively perceive the robot's surroundings in a stereoscopic manner. Additionally, the system mirrors the operator's arm and hand movements on the robot, creating an immersive experience as if the operator's mind is transmitted to a robot embodiment. We validate the effectiveness of our system by collecting data and training imitation learning policies on four long-horizon, precise tasks (*Can Sorting*, *Can Insertion*, *Folding*, and *Unloading*) for two different humanoid robots and deploy them in the real world.

---

[*]equal contribution

8th Conference on Robot Learning (CoRL 2024), Munich, Germany.

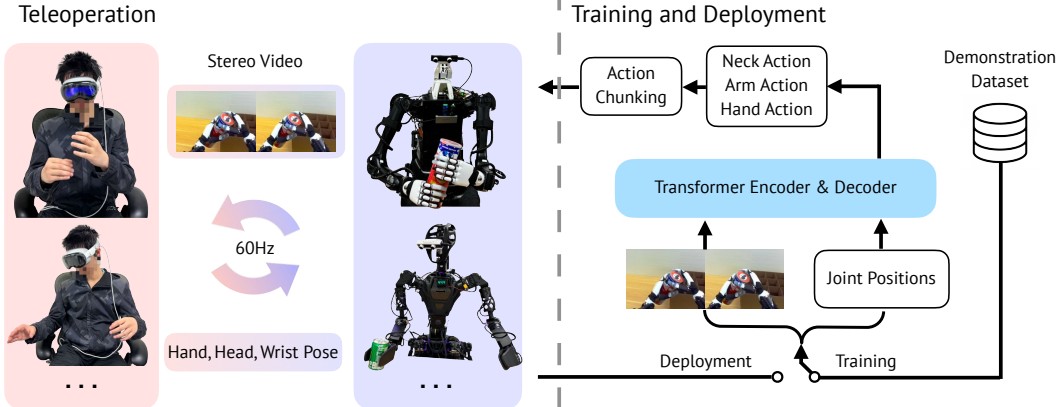

Figure 2: **Teleoperated data collection and learning setup.** Left: our teleoperation system. VR devices stream the hand, head, and wrist poses to the server. The server retargets the human poses to the robot and sends joint position targets to the robot. Right: we train an imitation policy for each task with action chunking transformer. The transformer encoder captures the relationship of image and proprioception tokens and the transformer decoder outputs action sequences of a certain chunk size.

# 1 Introduction

Learning-based robotic manipulation has advanced to a new level in the past few years, thanks to large-scale real-robot data [1, 2]. Teleoperation has been playing an important role in data collection for imitation learning, where it not only offers accurate and precise manipulation demonstrations, but also provides natural and smooth trajectories that allow the learned policies to generalize to new environment configurations and tasks. Various teleoperation approaches have been studied using VR devices [3, 2, 4, 5], RGB cameras [6, 7, 8], wearable gloves [9, 10, 11], and customized hardwares [12, 13].

There are two major components in most teleoperation systems: actuation and perception. For actuation, using joint copy to puppeteer the robot provides high control bandwidth and precision [12, 14, 15]. However, this requires the operators and the robot to be physically in the same location, not allowing for remote control. Each robot hardware needs to be coupled with one specific teleoperation hardware. Importantly, these systems are not able to operate multi-finger dexterous hands yet. For perception, the straightforward way is to observe the robot task space with the operator's own eyes in a third-person view [7, 6, 3] or a first-person view [16, 17]. This inevitably will cause occlusion on the operator's sight during teleoperation (e.g., occluded by robot arms or torso), and the operator cannot ensure the collected demonstration has captured the visual observation needed for policy learning. Importantly, for fine-grained manipulation tasks, it is hard for the teleoperator to look closely and intuitively at the object during manipulation. Displaying a third-person static camera view or using passthrough in the VR headset [3, 2, 18] encounter similar challenges.

In this paper, we propose to revisit teleoperation systems with VR devices. We introduce Open-TeleVision, a general framework to perform teleoperation with high precision applicable to different VR devices on different robots and manipulators. We experiment with two humanoid robots including Unitree H1 [19] humanoid robot with multi-finger hands and Fourier GR1 [20] humanoid robot with grippers, on bimanual manipulation tasks (Fig. 1). With the captured human operators' hand pose, we perform re-targeting to control multi-finger robot hands or parallel-jaw grippers. We rely on inverse kinematics to convert the operator's hand root position to the robot arm end-effector position.

Our major contribution to allowing fine-grained manipulations comes from perception, which incorporates VR systems with active visual feedback. We use a single active stereo RGB camera on the robot head equipped with two or three DoFs actuation, mimicking human head movement to observe a large workspace. During teleoperation, the camera moves along with the operator's head, streaming real-time, ego-centric 3D observations to the VR device. The human operator sees what

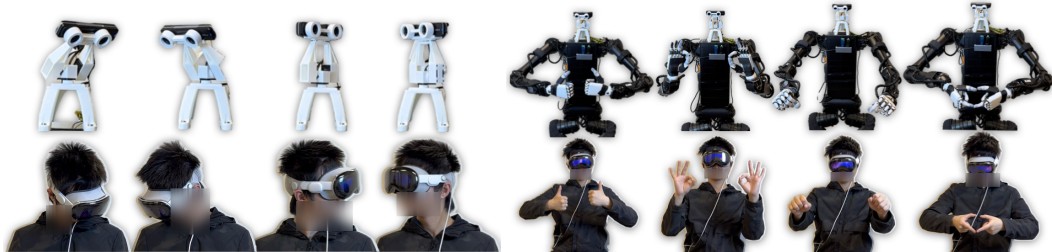

(a) Head movements.  (b) Arm and hand movements.

Figure 3: **Open-TeleVision enables high-DOF control of a head-mounted active camera and upper body movements.** Left: the robot's head follows the movement of the operator, providing intuitive spatial perception. Right: the robot's arm and hand movements are mapped from the operator with IK and motion retargeting.

the robot sees. This **first-person active sensing** brings benefits for both teleoperation and policy learning. For teleoperation, it provides a more intuitive mechanism for the user to explore a broader view when moving the robot's head, and attend to the important regions for detailed interactions. For imitation learning, our policy will imitate how to move the robot head actively together with manipulation. Instead of taking a further and static captured view as inputs, active camera provides a natural attention mechanism to focus on next-step manipulation related regions and reduce the pixels to process, allowing smooth, real-time, and precise close-loop control. Another important innovation in perception is **streaming stereoscopic video** from the robot view to human eyes. The operator can have a better spatial understanding which is shown to be crucial for completing tasks in our experiments. We also show training with stereo image frames improves the performance of the policy.

Our experiments follow the teleoperation and imitation learning paradigm. We experiment with multiple fine-grained manipulation tasks across two robots as shown in Fig. 1. For teleoperation, we qualitatively show the benefits of using an active camera allows more intuitive and focused observation, and quantitatively show streaming stereoscopic video allows better success rate and shorter completion time across different users. For imitation learning, we quantitatively report active camera sensing enables faster inference for real-time smooth control, and using stereoscopic inputs achieves better manipulation performance. Our policy can conduct **long-horizon** tasks such as inserting multiple cans in a sequence. A key benefit of our system is enabling **remote control** by an operator via the Internet.

## 2 TeleVision System

Our system overview is shown in Fig. 2. We develop a web server based on Vuer [21]. The VR devices stream the operator's hand, head and wrist poses in $SE(3)$ to the server, which handles the human-to-robot motion retargeting. Fig. 3 shows how the robot's head, arm, hand follows the human operators movements. In turn, the robot streams stereo video at a resolution of 480x640 for each eye. The entire loop happens at 60 Hz. While our system is agnostic to VR device model, we choose Apple VisionPro as our VR device platform in this paper.

**Hardwares:** For robots we choose two humanoid robots as shown in Fig. 4: Unitree H1 [19] and Fourier GR-1 [20]. We only consider their active sensing neck, two seven-DoF arms and end-effectors, while other DoFs are not used. H1 has six DoFs for each hand from [22], while GR-1 has a one-DoF jaw gripper. For active sensing, we design a gimbal with two revolute DoFs (yaw and pitch) mounted on top of the torso of H1. The gimbal is assembled from 3D printed parts and powered by DYNAMIXEL XL330-M288-T motors [23]. For GR-1, we use the three-DoF neck (yaw, roll, and pitch) that comes with the manufacturer. A ZED Mini [24] stereo camera is used with both robots to provide stereo RGB streaming. We mostly feature humanoid robots in our setup because the teleoperation issue of occlusion and lack of intuitiveness of current system is the most prominent. While our system is specifically tailored for humanoid robots to maximize the capability

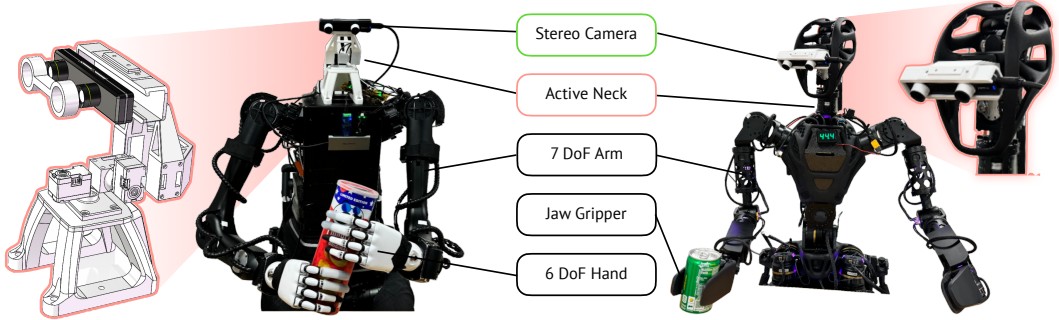

Figure 4: **Reference design of Open-TeleVision on two types of hardware.** Left: Unitree H1 [19] with six-DoF Inspire [22] hands. The head contains yaw and pitch motors. Right: Fourier GR-1 [20] with jaw gripper. The active neck is from the manufacturer with yaw roll and pitch motors.

for immersive teleoperation experiences, it is versatile enough to be applied to any setup featuring two arms and one camera.

## 3 Experiments

In this section, we aim to answer the following questions:

- How do the key design choices of our system affect the performance of imitation learning results?
- How effective is our teleoperation system in collecting data?

We choose ACT [12] as our imitation learning algorithm with two key modifications. First, we replace the ResNet with a more powerful visual backbone DinoV2 [25, 26], a pretrained vision transformer (ViT) by self-supervised learning. Second, we use two stereo images instead of four images from individually arranged RGB cameras as the input to the transformer encoder. The DinoV2 backbone produces $16 \times 22$ tokens for each image. The state token is projected from the current joint position of the robots. We use absolute joint positions as the action space. For H1 the action dimension is 28 (7 for each arm, 6 for each hand, and 2 for active neck). For GR-1 the action dimension is 19 (7 for each arm, 1 for each gripper, 3 for active neck). The proprioception token is projected from the corresponding joint position readings.

We choose four tasks with an emphasis on precision, generalization and long horizon to show the effectiveness of our proposed teleoperation system in Fig. 10. These tasks require actively looking to the left or right of the workspace using a single active camera, otherwise a multi-camera setup on the table. They also involve significant object location randomization and different manipulation strategies. We further show the data collection intuitiveness and speed by conducting user studies on all four tasks and comparing them with humans doing the same tasks.

### 3.1 Imitation Learning Results

We ablate our key design choices of the system and show from real-world experiments their effectiveness. The two baselines are *w. ResNet18*, which uses the original ACT visual backbone ImageNet Pre-trained ResNet18 [27], and *w/o Stereo Input*, which only takes the visual tokens from the left image instead of both. All models are trained using AdamW optimizer [28, 29] with a learning rate of $5e-5$, a batch size of $45$ and for $25k$ iterations on a single RTX 4090 GPU. We conduct *Can Sorting* task on both H1 and GR-1 robots. All other tasks are conducted with H1 robot only.

From our experiment results shown in Tab. 1, we notice that the original ResNet backbone (ImageNet pre-trained) fails to adequately perform all four tasks. In the original ACT implementation, four cameras are used (two fixed, two wrist-mounted) to alleviate the deficiency of explicit spatial information from using RGB images. Our setup involves only two stereo RGB cameras, potentially making spatial information retrieval more challenging for ResNet backbone.

| Baselines | H1 Can Sorting | | GR-1 Can Sorting | | Can Insertion | |
|---|---|---|---|---|---|---|
| | Pick | Place | Pick | Place | Pick | Insert |
| w. DinoV2 (Ours) | **92**% | **88**% | **87**% | 60% | **90**% | **87**% |
| w. ResNet18 | 74% | 58% | 83% | 50% | 53% | 70% |
| w/o Stereo Input | 46% | 52% | 73% | **63**% | 47% | 63% |

| Baselines | Folding | | | | Unloading | | |
|---|---|---|---|---|---|---|---|
| | Lift | Fold | Move | Fold | Extract | Pass | Place |
| w. DinoV2 (Ours) | **100**% | **100**% | **100**% | **100**% | **100**% | **100**% | **100**% |
| w. ResNet18 | 100% | 100% | 100% | 100% | 85% | 100% | 95% |
| w/o Stereo Input | 100% | 100% | 60% | 100% | 70% | 95% | 100% |

Table 1: **Success rate of autonomous policy.** We record five real-world episodes for each task. Each episode contains a complete task cycle defined in Sec. 3. On GR-1, each *Can Sorting* roll-out contains six pickings and six placings, accumulating 30 trials for each sub-task.

**Can Sorting:** We evaluate the success rate of picking up the can and the accuracy of placing it in the designated case separately. According to the results on H1 in Tab. 1, our model has the highest success rate in both evaluation metrics. *w. ResNet18* is distinctly inferior in both picking and sorting, likely due to its backbone's limitations. Without implicit depth information from stereo inputs, *w/o Stereo Input* fails to properly pick up the can (23/50 success rate). Its low sorting accuracy (26/50 success rate) is highly correlated to its poor performance in the previous stage, as an experimenter must frequently help it to grasp the can, an action that interferes with visual inference.

The results of *Can Sorting* with GR-1 are also reported in Tab. 1. In the picking sub-task, our model consistently outperforms the other two baselines. However, in the placing sub-task, none of the models reach satisfactory accuracy. This phenomenon can be attributed to the different morphologies between the dexterous hand and the gripper: when a robot hand grasps the can, the camera is able to see the color of the can and make its actions based on what it sees; in contrast, when a gripper grasps the can, the camera can barely make out the color due to significant occlusion(Fig. 10b), complicating visual inference. The other limitation is that ACT's ability to make long-horizon inference is dependent on its chunk size. In our setup, using a chunk size of 60 with an inference frequency of 60 Hz effectively provides the robot with one second of memory: when the robot is supposed to drop the can without direct visual confirmation, it has likely forgotten the color of the can which was visible at the time of pickup.

**Can Insertion:** We evaluate the success rate of picking up and insertion separately. The results (Tab. 1) suggest that our model surpasses the baseline models in both metrics. Successfully pinching up the soda can with only two fingers and adjusting its pose to fit into the slot requires precise control underpinned by spatial reasoning. This capability is notably absent in *w/o Stereo Input*.

**Folding:** We evaluate the success rate based on the policy's ability to perform two consecutive folds without dropping the towel. The results are reported in Tab. 1. Both ours and *w. ResNet18* models achieve a 100% success rate in performing the folds. We speculate that the high success rate of this task across all models is due to its high repetitiveness in actions. On the other hand, *w/o Stereo Input* occasionally fails (2/5 fails) at the third stage, in which the robot should gently move the towel to the edge of the table to ease the second fold. We observe that *w/o Stereo Input* tends to press its hands too hard on the table and prohibit the successful movement of the towel. This action is likely due to the lack of depth information from using a single RGB image, as this step requires the robot to adjust the force exerted by its hands based on the distance to the table.

**Unloading:** We evaluate the success rate on three consecutive stages individually: extracting the tube, in-hand passing, and placing. As shown in Tab. 1, our model reaches 100% success rate in all three stages. Both *w. ResNet18* and *w/o Stereo Input* fail at extracting the tube from certain slots (3/20 fails and 6/20 fails, respectively). The extraction is particularly hard for *w/o Stereo Input*, partly because it

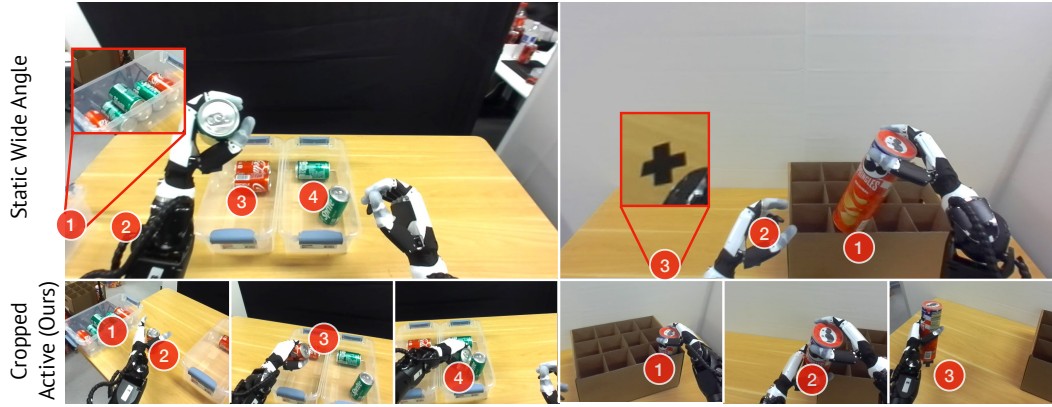

Figure 6: **Comparison between wide-angle lenses and cropped views.** Left: H1 *Can Sorting*. Right: *Unloading*. Wide angle images are of resolution 1280x720 and have a 102°(H) x 57°(V) field-of-view. Cropped images are cropped from the bottom middle from the original wide-angle images, and are of resolution 640x480. Red marks are the points of interests (PoIs) for the tasks. For *Can Sorting*, the PoIs are the bin with cans to be sorted, pick location, coke bin and sprite bin. For *Unloading*, the PoIs are tube slot, in-hand passing, and the drop location.

cannot estimate the relative orientation between the tube and the hand correctly. In-hand passing and placing are relatively simpler because these two stages do not involve much randomization during data collection. Nonetheless, *w. ResNet18* and *w/o Stereo Input* still fail at them occasionally (both 1/20 fails) while our model completes these two stages with no mistake.

## 3.2 Generalization

We evaluate the generalization capabilities of our model under randomized conditions. In the *Can Sorting* task conducted with H1, we assess its success rate of picking from a 4x4 grid with each cell measuring 3 cm, as depicted in Fig.5 (left). The results are detailed in Fig.5 (right), which indicate that our policy generalizes well to large areas covered in the dataset, maintaining a 100% success rate. Even in the peripheral regions, which are rare or absent in the demonstrations, our model still exhibits adaptability to complete the grasp at times.

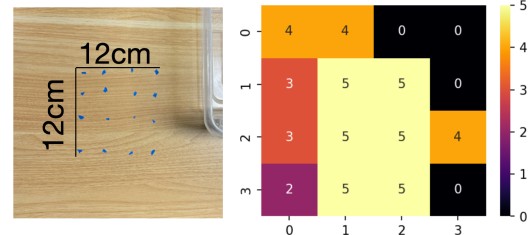

Figure 5: **Distribution of can placements.** Left: The cans are placed in a f $4 \times 4$ grid. Right: number of successful pickings heatmap with five trials at each location.

**Why Use Active Sensing?** We compare the view from a wide-angle camera and our active camera setup with a cropped view in Fig. 6. A single static wide-angle camera still have trouble capturing all of the points-of-interest (PoI). One has to mount multiple cameras [12, 14] or tune the camera positions for each task. The static wide angle camera also captures non-relevant information that brings additional computation cost for both training and deployment as shown in Tab. 2. Our $TeleVision$ system is 2x faster for training with the same batch size and can accommodate 4x data in one batch on a 4090 GPU. During inference, our system is also 2x faster, leaving sufficient time for IK and retargeting computation to reach 60Hz deployment control frequency. When using wide angle images as input, the inference speed is lower, at about 42 frames per second.

Furthermore, with a static camera, the operator needs to gaze at a PoI on the edge of the image, which brings additional discomfort and non-intuitiveness as humans use central (foveal) vision to focus [30, 31].

**User Study.** We validate our design choice of streaming and displaying stereo images in the VR headset through a user study. The use study is performed on four participants with varied levels of

| Method | Average Training Batch Time (s) | Average Deploying Step Time (s) |
|---|---|---|
| Cropped Active 45 (Ours) | 0.41 | **0.012 (83Hz)** |
| Cropped Active 10 | **0.10** | - |
| Wide Angle 10 | 0.32 | 0.024 (42Hz) |

Table 2: **Computation cost comparisons between the active vs non-active vision setup.** We sample 100 batches for training and 100 deployment steps and average their computation time. The number in the baseline name indicates batch size. *Cropped Active* is down-sampled from 640x480 to 308x224, resulting in 352 image tokens for each camera. *Wide Angle* is down-sampled with the same scale to 588x336, resulting in 1008 image tokens for each camera. We use batch size 10 for *Wide Angle* due to the memory limit of an RTX 4090 GPU.

| User | Stereo | | | | Mono | | | |
|---|---|---|---|---|---|---|---|---|
| | Can Sorting | Can Insertion | Folding | Unloading | Can Sorting | Can Insertion | Folding | Unloading |
| #1 | 67 | 54 | 30 | 73 | 110 | 116 | 49 | 94 |
| #2 | 59 | 55 | 44 | 87 | 91 | 92 | 67 | 119 |
| #3 | 55 | 59 | 27 | 52 | 76 | 68 | 37 | 96 |
| #4 | 82 | 62 | 36 | 55 | 88 | 62 | 47 | 78 |
| Mean | **66** | **58** | **34** | **67** | 91 | 85 | 50 | 97 |

(a) Completion time for individual participant in second.

| User | Stereo | | | | Mono | | | |
|---|---|---|---|---|---|---|---|---|
| | Can Sorting | Can Insertion | Folding | Unloading | Can Sorting | Can Insertion | Folding | Unloading |
| #1 | 100% | 100% | 100% | 100% | 100% | 33% | 100% | 50% |
| #2 | 100% | 100% | 100% | 100% | 100% | 83% | 100% | 50% |
| #3 | 100% | 100% | 100% | 100% | 90% | 83% | 100% | 50% |
| #4 | 100% | 100% | 100% | 100% | 80% | 83% | 100% | 50% |
| Mean | **100**% | **100**% | **100**% | **100**% | 93% | 71% | 100% | 50% |

(b) Success rate for individual participant.

Table 3: **User study results.** Users see stereo video stream in *Stereo* and see a single RGB video from the left camera in *Mono*.

prior exposure to VR devices. The participants are graduate students aged 20-25. Each participant is asked to complete all four tasks under guidance and is given roughly five minutes to familiarize with the system and the tasks. Their performances in both stereo and monocular (*Mono*) setup are detailed in Tab. 3. Both metrics (task completion time and success rate) suggest that *Stereo* surpasses *Mono* by a large margin. Additionally, based on qualitative results of user feedback, using stereo image is remarkably better than using a single RGB image. Typically, human eyes are adaptive enough to infer depth and spatial relationships from a single RGB image. However, in teleoperation cases where the operator must actively interact with different objects, such intuitions are often proven insufficient. Leveraging the spatial information available in stereo images can substantially alleviate the discomfort when teleoperating robots with only images.

## 4 Related Work

**Data Collection.** Learning based methods have achieved great success in locomotion control with massive simulation data and Sim2Real transfer [32, 33, 34, 35, 36, 37, 38, 39, 40, 41, 42, 43, 44]. On the other hand, collecting real-robot demonstrations for imitation learning has shown to be a more

effective way for robotic manipulation [45, 46, 1, 47, 48]. This is mainly due to the large Sim2Real gap with complex contacts between the manipulator and objects and surroundings. To collect real-world data, teleoperation has emerged as the mainstream approach using RGB cameras [6, 7, 8], mocap gloves [9, 10, 11], and VR devices [3, 2]. There are also more conventional frameworks exploiting exo-skeleton devices [13] or mirroring arms for joint copy [12, 14, 15]. For example, the recent ALOHA framework [12] provides precise control on fine-grained manipulation tasks with exact joint mapping. In this paper, instead of adopting joint copy, we find VR-based teleoperation system with hand retargeting can also achieve precise control of fine-grained manipulation tasks. Tab. 7 in the supplementary material summarizes the difference between our system and previous teleoperation systems.

**Bimanual Teleoperation.** Access to an intuitive and responsive teleoperation system is essential for collecting high-fidelity demonstrations. Most existing teleoperation systems are restricted to using grippers [13, 49, 15, 16] or single-hand setups [50, 51, 52], which tend to be either non-intuitive or limited in their capabilities. We believe enabling manipulation with multi-finger hands in 3D space allows more robust manipulation conducted on diverse tasks, providing significant advantages over existing teleoperation systems. For example, it will be very challenging for a parallel-jaw gripper to stably grasp a Pringles Chips tube. For the handful of bimanual-hands setups [3, 2], they require operators to control the robot by directly observing its hands during task execution or seeing an RGB image from a fixed camera. In contrast, our system integrates a first-person stereo display with active head rotations, offering an intuitive interface as if the robot is an avatar of the operator itself. Our system allows the operator to control the robot remotely far away (e.g., control the robot on the West Coast from the East Coast). This is also the major innovation that differentiates our work from concurrent works on humanoid teleoperation [53, 54]. Our system allows smooth teleoperation on fine-grained tasks, and precise control policy for long-horizon execution. Furthermore, our web-based system is accessible from all kinds of portable devices such as Quest (∼500 USD) or VisionPro(∼3500 USD), while [55, 56, 57] require dedicated teleoperation stations that can be very expensive. For example, The two Franka Emika Panda arms in [55] will cost approximately 20000 USD, excluding other components of the station and limit the mobility of the system (operators must come to the station to start operating).

**Imitation learning.** The topic of imitation learning has covered a wide range of literature. If we distinguish the existing work with sources of demonstrations, we can classify them into learning from real-robot expert data [16, 12, 14, 58, 59, 60, 61, 46, 62, 1, 63, 47, 64, 65], learning from play data [66, 67, 68], and learning from human demonstrations [67, 69, 70, 71, 72, 73, 74, 52]. Besides learning for manipulation, motion imitation for physical characters and real robots have also been widely studied in [75, 76, 36, 77, 78, 79, 80, 81, 82, 80, 83, 84, 85, 86, 87, 88]. This paper falls into imitation learning for manipulation tasks using real-robot data collected by human teleoperation.

## 5   Conclusion and Limitations

In this paper, we propose a system for immersive teleoperation with stereoscopic video streaming and active perception with actuated necks. We show that our system can enable precise and long-horizon manipulation tasks and the data collected with our system can be readily utilized by imitation learning algorithms. We also conduct user studies to show the importance of stereo perception for human operators. While the stereo perception is crucial for the operators' spatial understanding, there is still a lack of other forms of feedback, such as haptic feedback, which is typically the dominant feedback with first-person visual occlusion and in tactile-intensive tasks. A system that enables the relabeling of expert data could be very helpful for increasing success rate, which is also missing from our system now. The future work can be extended to mobile version which utilizes all the DoFs of the robot.

**Acknowledgments**

This work was supported, in part, by Amazon Research Award and the Intel Rising Star Faculty Award.

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

# A    Arm and Dexterous Hand Control

**Arm Control:** The human wrist poses are first converted into the robot's coordinate frame. Specifically, the relative positions between the robot end-effectors and the robot head are expected to match those between the human wrists and head. The orientations of the robot wrists are aligned with the absolute orientations of the human wrists, as estimated during the initialization of the Apple VisionPro hand tracking backend. This differentiated treatment of end-effectors' positions and orientations ensures the stability of the robot end-effectors when the robot's head moves along with human's head. we employ the Closed-loop Inverse Kinematics (CLIK) algorithm based on Pinocchio [89, 90, 91] to compute the joint angles of the robot's arm. The input end-effector poses are smoothed using an $SE(3)$ group filter, implemented with Pinocchio's $SE(3)$ interpolation, enhancing the stability of the IK algorithm. To further mitigate the risk of IK failures, our implementation incorporates a joint angle offset when the arm's manipulability approaches its limits. This correction procedure has minimal impact on end-effector tracking performance, as the offset is projected onto the nullspace of the robot arm's Jacobian matrix, thereby preserving tracking accuracy while addressing the constraints.

**Hand Control:** The human hand keypoints are translated into robot joint angle commands through dex-retargeting, a highly versatile and fast-computing motion retargeting library [6]. Our approach utilizes vector optimizers on both dexterous hand and gripper morphologies. The vector optimizers formulate the retargeting problem as an optimization problem [6, 92] while the optimization is defined based on user-selected vectors:

$$\min_{q_t} \sum_{i=0}^{N} ||\alpha v_t^i - f_i(q_t)||^2 + \beta ||q_t - q_{t-1}||^2. \tag{1}$$

In the above formulation, $q_t$ denotes the robot joint angles at time $t$, and $v_t^i$ is the i-th keypoint vector on the human hand. The function $f_i(q_t)$ computes i-th keypoint vector on the robot hand using forward kinematics from the joint angles $q_t$. The parameter $\alpha$ is a scaling factor that accounts for the hand size difference between the human hand and the robot hand (we set it as 1.1 for Inspire hands). The parameter $\beta$ weights the penalty term that ensures temporal consistency between consecutive steps. The optimization is conducted in real-time using Sequential Least-Squares Quadratic Programming (SLSQP) algorithm [93] as implemented in NLopt library [94]. The computation of forward kinematics and its derivatives are conducted in Pinocchio [90].

For dexterous hands, we employ seven vectors to synchronize the human and robot hands: five vectors represent the relative positions between the wrist and each fingertip keypoint; two additional vectors, spanning from the thumb fingertip to the primary fingertips (index and middle), are incorporated to enhance the motion accuracy during fine-grained tasks. For grippers, optimization is achieved using a single vector, defined between the human thumb and index fingertips. This vector is aligned with the relative position between the gripper's upper and lower ends, enabling intuitive control over the grippers opening and closing motions by simply pinching the operator's index and thumb fingers.

# B    Discussion on Comparing with Prior Teleoperation Systems

We discuss from two critical perspectives of teleoperation: actuation and perception.

**Actuation.** Various approaches have been studied for teleoperating robots through human commands, including visual tracking, motion-capture devices, and joint copying through customized hardware. While using motion-capture gloves for teleoperation seems the most intuitive, the commercially available gloves are not only costly but also unable to provide wrist pose estimations. The joint copying method has drawn significant attention recently, following the success of ALOHA[12]. This method offers precise and dexterous control. Historically, this method was considered costly, requiring using an additional pair of identical robotic arms for teleoperation; nonetheless, this issue has been mitigated by the adoption of low-cost exoskeleton devices to transmit commands[13]. Despite their simplicity, joint copying systems are currently limited to using grippers and have not yet been extended to operate multi-finger hands. Conversely, visual tracking employs off-the-shelf

hand pose extractors to track finger movements, but relying solely on RGB or RGBD images can lead to noisy and imprecise data. The recent surge in VR technology has led to the development of teleoperation systems that utilize VR tracking. VR headset manufacturers often integrate built-in hand-tracking algorithms that fuse data from diverse types of sensors, including multiple cameras, depth sensors, and IMUs. Hand-tracking data collected through VR devices are generally considered more stable and accurate than self-developed vision-tracking systems, while the latter only utilize a subset of the mentioned sensors (RGB+RGBD[6], Depth+IMU[8], etc.).

**Perception.** While being the other critical component of teleoperation, perception has been considerably less explored than actuation within this field. Most existing teleoperation systems require the operators to directly observe the robot's hands using their own eyes. While direct viewing provides the operators with depth sensing, leveraging humans' inherent capability for stereoscopic vision, it restricts the system to be non-remote, necessitating the physical presence of the operator. Some teleoperation systems circumvent this by streaming RGB images, enabling remote control[6, 7]. However, if the operator opts for remote controlling by watching an RGB stream, the benefits of depth sensing provided by the human eye are lost. Despite being capable to provide both remote controlling and depth sensing, these two features are mutually exclusive in these systems. OPEN TEACH[2] merges the two in a mixed-reality fashion, yet it still requires the operator to be in proximity to the robot, otherwise the depth sensing is unavailable. Prior to Open-TeleVision, no system offered both remote control and depth sensing simultaneously: the operator is forced to choose between either direct viewing, which demands physical presence, or RGB streaming, which abandons depth information. By utilizing stereo streaming, our system is the first to provide both functionalities within a single setup.

## C  Discussion of Visual Occlusion

To support our proposed assumption that the unsatisfactory performance observed in *GR-1 Can Sorting* task stems from visual occlusion caused by GR-1's gripper end-effector, we perform a controlled experiment. In the new experiment, we add color labels to the cans to mitigate the occlusion factor, as depicted in Fig. 7 left. The other settings are identical to those described in the *GR-1 Can Sorting* task in the article. Results are recorded in Table. 4.

| Baselines | GR-1 Can Sorting | | | |
|---|---|---|---|---|
| | Pick(new) | Pick(old) | Place(new) | Place(old) |
| w. DinoV2 (Ours) | 97% | 87% | 100% | 60% |
| w. ResNet18 | 90% | 83% | 97% | 50% |
| w/o Stereo Input | 47% | 73% | 93% | 63% |

Table 4: **Success rate for *GR-1 Can Sorting*.** The experiments are conducted under identical settings and number of trials as outlined in Tab. 1. Columns marked as (old) contain the original results using unlabeled cans, while the columns marked as (new) contain the results of the new experiment using labeled cans.

The results indicate a substantial improvement in the success rate of the placing task across all three baselines, achieved by using labeled cans. Our model reach a 100% accuracy rate in the placement, compared to the previous 0.60; notable gains are also observed in the other baselines, with *w. ResNet18* improving from 0.50 to 0.97, and *w/o Stereo Input* improving from 0.63 to 0.93. On the other hand, while success rates of picking also increase for our model and *w. ResNet18*, *w/o Stereo Input* does not exhibit similar improvements. This disparity further validates our claim that a successful can-picking requires spatial information from stereo images.

As with H1 in Sec. 3.2, we perform an experiment to evaluate the model's generalization capability with *Can Sorting* on GR-1 with labeled cans. Its results are similarly collected from a 4x4 grid (the same as Fig. 5 left) with each cell measuring 3 cm. Generalization results are shown in the heatmap in Fig. 7 right.

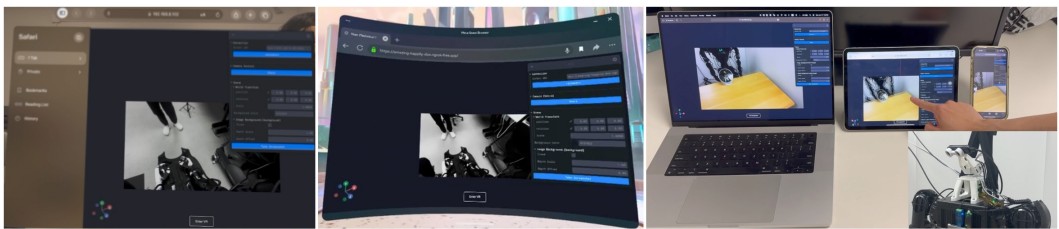

Figure 9: **Our web-based cross-platform system enables access across different devices.** Left: Apple Vision Pro. Middle: Meta Quest. Right: Macbook, iPad and iPhone. On VR devices, the users can enter an immersive session to start teleoperation with hand and wrist pose streaming. On other devices, hand and wrist streaming are not available but the user can still see the streamed images and control the robot's active neck by dragging on the devices' screen.

The results suggest that our model can easily adapt to most of the random locations covered in our experiment, reaching 100% grasping accuracy in nearly all locations on the grid. The results as shown here for *GR-1 Can Sorting* are also notably better than the results as shown in Fig. 5 for *H1 Can Sorting*. The difference may also be attributed to differences in end-effector morphologies. Grasping a soda can, which requires less dexterity and more tolerance, is better suited to grippers than to robotic hands.

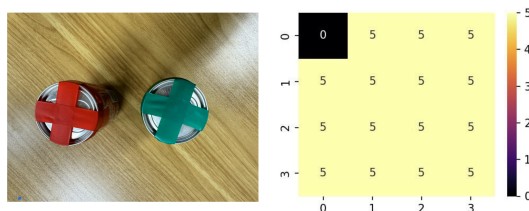

Figure 7: **Can labeling and generalization results.** Left: Figure depicting labeled cans. Right: number of successful pickings heatmap with five trials at each location.

For H1 robot's setup, the anthropomorphic hands we use are provided by Inspire Robots [22]. A close-up of one of the Inspire Hands is shown in Fig. 8.

## D   Dexterous Hand

Each hand has five fingers and twelve DoFs, among which six are actuated DoFs: two actuated DoFs are on the thumb and one on each of the remaining fingers. Each non-thumb finger possesses a single actuated revolute joint at the metacarpophalangeal (MCP) joint, serving as the entire finger's actuating DoF. The proximal interphalangeal (PIP) joints of these four fingers are driven by the MCP joints through linkage mechanisms, adding four underactuated DoFs. The thumb is equipped with two actuated DoFs at the carpometacarpal (CMC) joint. The thumb's MCP and interphalangeal (IP) joints are also driven by linkage mechanisms, contributing to additional two underactuated DoFs.

## E   Teleoperation Interface

Figure 8: **Inspire Hand [22].**

Fig. 9 shows our web-based cross-platform interface that can be accessed not only from VR devices but also laptops, tablets and phones.

## F   Experimental Details and Hyperparameters

### F.1   Autonomous Task Description

**Can Sorting:** This task as shown in Fig. 10a involves sorting randomly placed Coke cans (red) and Sprite cans (green) on a table. The cans are placed on the table one-by-one but with random positions and types (Coke or Sprite). The goal is to pick up each can on the table and toss it into the designated

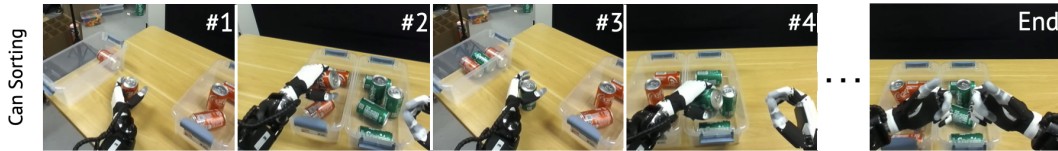

(a) The robot picks up a randomly placed can (*#1*), places the red can in the left case (*#2*), picks up the next can (*#3*), places the green can in the right case (*#4*). When the sorting box at the edge of the table is empty (as shown in *#1, #3*) and there is no more can on the table, the robot poses the ending gesture (*End*).

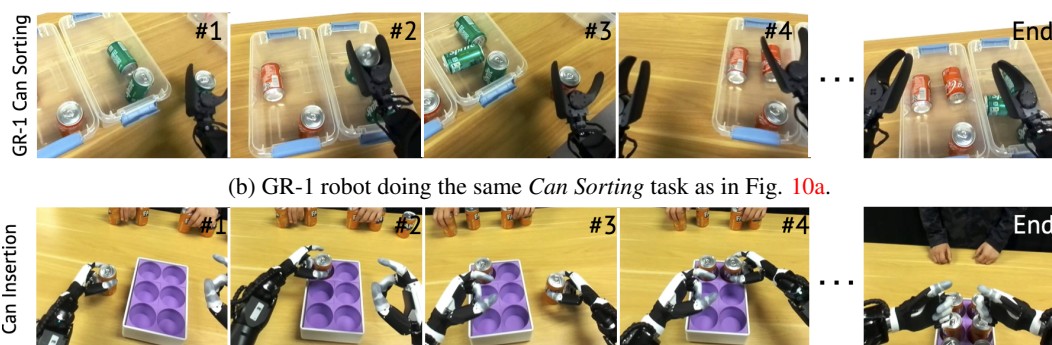

(b) GR-1 robot doing the same *Can Sorting* task as in Fig. 10a.

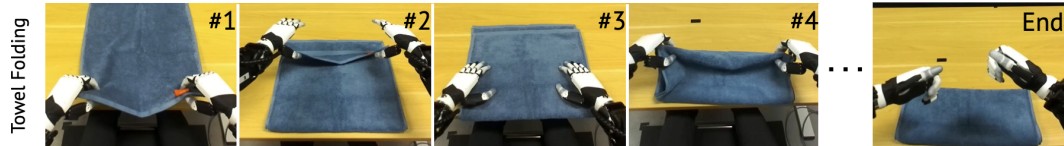

(c) The robot picks up the can (*#1*), inserts the can into the slot (*#2*), uses the other hand to repeat picking and inserting(*#3, #4*). When all six slots have been filled, the robot poses the ending gesture (*End*).

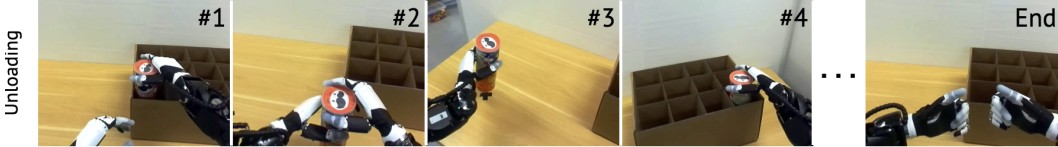

(d) The robot picks up the two corners of the towel (*#1*), folds the towel (*#2*), gently moves the towel to the edge of the table (*#3*), folds the towel twice (*#4*). When the towel has been folded into a satisfactory configuration, the robot poses the ending gesture (*End*).

(e) The robot extracts the tube from a random slot (*#1, #4*), passes from right hand to left (*#2*), places the tube in the designated position (*#3*). When there is not a tube anymore, the robot poses the ending gesture (*End*).

Figure 10: **Data collection using H1 on four tasks.** Each row represents one task. At the end of each task, the operator postures to the same ending gesture to signify the successful completion of one demonstration. Then our system will stop recording data. We do not crop the ending gesture from our dataset. The learned policy can successfully react to ending conditions.

case: left for Coke and right for Sprite. Solving this task requires the robot to adaptively generalize upon the position and orientation of each can for accurate grasping. It also demands that the policy can adjust its planned motions based on the color of the can it is currently holding. Each episode consists of sorting ten cans (five Sprite five Coke randomly) consecutively. **Can Insertion:** This task shown in Fig. 10c involves picking up soft drink cans from the table and carefully inserting them into slots within a container in a predefined sequence. While both involve manipulation of drink cans, this task demands more precise and fine-grained actions than the previous one as a successful insertion necessitates high accuracy. In addition, a different grasping strategy is adopted in this task. In the previous can-sorting task, the robot only needs to toss the can into a designated area, hence we form a grasp that involves the palm and all five fingers, which is a tolerant but imprecise grasping strategy. In this one, to insert the can into a slot that is only slightly larger than the can (the diameter of the soda can is roughly 5.6 cm, and the diameter of the slot is roughly 7.6 cm), we employed a

more pinching-like strategy that utilizes only the thumb and index fingers, enabling more granular adjustments in the placement of the cans. The two distinct grasping strategies demonstrate that our system is able to accomplish tasks with complex hand gesture requirements. Each episode of this task includes picking and placing all six cans into the right slots.

**Folding:** This task shown in Fig. 10d involves folding the towel twice. The distinction of task is that it manifests the system's capability to manipulate soft and compliant materials like a towel. The action sequence of this task unrolls as follows: pinching the two corners of the towel; lifting and folding; gently moving the towel to the edge of the table to prepare for a second fold; pinching, lifting, and folding again. Each episode of this task consists of one complete folding of the towel.

**Unloading:** This task shown in Fig. 10e is a composite operation that involves tube extraction followed by in-hand passing. In this task, a chip tube is randomly placed into one the four slots within a sorting box. The goal is to identify the slot containing the tube, extract the tube using the right hand, pass it to the left hand and place the tube in a predefined location. To successfully execute this task, the robot needs both visual reasoning to discern the tube's location and accurate action coordination for extraction and in-hand passing. Each episode of this task consists of picking up four tubes from four random slots, passing them to another hand, and finally putting them on the table.

## F.2 Experimental Details

| Tasks | Average Episode Length (s) | Number of Episodes |
|---|---|---|
| H1 Can Sorting | 93±5 | 10 |
| GR-1 Can Sorting | 61±5 | 10 |
| Can Insertion | 84±7 | 20 |
| Folding | 44±5 | 20 |
| Unloading | 93±6 | 20 |

Table 5: **Details about collected demonstration data for each task.**

More experimental details are listed in Tab. 5. All tasks, with the exception of *Can Sorting* (both *H1 Can Sorting* and *GR-1 Can Sorting*), use 20 human demonstrations for training. In contrast, only 10 demonstrations are used for *Can Sorting*. This choice is primarily due to its repetitive nature: each episode consists of 10 (6 for *GR-1 Can Sorting*) individual can-sortings. Consequently, 10 demonstrations encompass 100 individual sorting rollouts, providing ample data for training.

## F.3 Hyperparameters

The hyperparameters employed for training the ACT [12] models are detailed in Table. 6. While the majority of these hyperparameters are consistent across all baselines and all tasks, there are a few exceptions, including chunk size and temporal weighting. The detailed explanations are as follows.

| | |
|---|---|
| KL weight | 10 |
| chunk size | 60 |
| hidden dimension | 512 |
| batch size | 45 |
| feedforward dimension | 3200 |
| epochs | 25000 |
| learning rate | 5e-5 |
| temporal weighting | 0.01 |

Table 6: **Hyperparameters of ACT.**

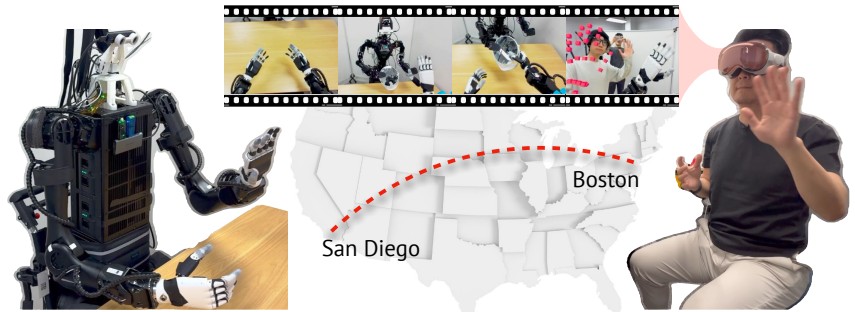

Figure 11: **Our system enables cross-country teleoperation over the internet.** The operator at Boston, USA can operate a robot at San Diego, USA (approximately 3000 miles away). The robot and operator pictures are flipped left to right for better illustration.

The definition of chunk size in the action chunking operation is outlined in the original ACT paper[12]. We use a chunk size of 60 for all tasks, with the exception of *Can Insertion*, in which we use a chunk size of 100. Using a chunk size of 60 in our setup effectively provides the robot with approximately one second of memory, correlating with our inference and action frequency of 60Hz. Nonetheless, we notice that in *Can Insertion* task, using a larger chunk size, which corresponds to incorporating more historical actions, proves to be advantageous for the model to perform correct action sequences.

The definition of temporal weighting in the temporal aggregation operation is outlined in the original ACT paper[12], where an exponential weighting scheme $w_i = exp(-m * i)$ is employed to assign weights to actions at different timesteps. $w_0$ is the weight for the oldest action, adhering to ACT's setting. $m$ is the temporal weighting hyperparameter mentioned in Table. 6. As $m$ decreases, greater emphasis is placed on more recent actions, rendering the model more reactive but less steady. We found that using a temporal weight $m$ of 0.01 reaches a satisfactory balance between responsiveness and stability for most tasks. However, for *Unloading* and *Can Sorting* tasks, we adjust this parameter to cater to their specific needs. For unloading, $m$ is set as 0.05, ensuring greater stability during in-hand passing; for *Can Sorting*, $m$ is set as 0.005, providing quicker movements.

**Teleoperated Performance.** In Fig. 12, we include more teleoperation tasks that our system is capable of. The *Wood-board Drilling* task shows that our system can operate heavy-weight ($1kg$) tools that are designed for humans, thanks to its compatibility with dexterous hands, and can apply sufficient force to the wood board to drill it through. Such a task is virtually impossible for the grippers. The *Earplugs Packing* task demonstrates that our system is dexterous and responsive enough to perform agile bimanual arm-hand coordination. The *Pipette* task demonstrates that our system is also capable of precise actions. This is also a task that is extremely hard or impossible for the grippers to achieve, as the usage of a pipette is specialized for anthropomorphic hands. Even though the motors on H1 humanoid robot are quasi-direct-drive motors with planetary reducers, which are known to have gear clearance and far less accuracy and stiffness, our system can still achieve high-precision with human operators in the loop. Our system also achieves remote teleoperation as shown in Fig. 11. Please see our videos for visualization.

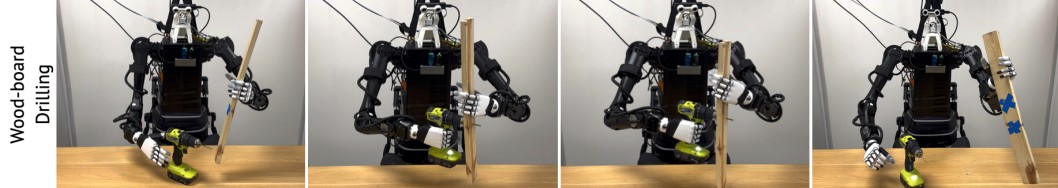

(a) The robot holds a wood board of thickness $2cm$ with the left hand and uses an electric drill to drill 2 holes on the board. This task requires precise control of the drill trigger using the index finger. Furthermore, our system enables fine control of the hand so that after drilling the first hole, the robot can let the board slide in hand to leave space for the second drilling.

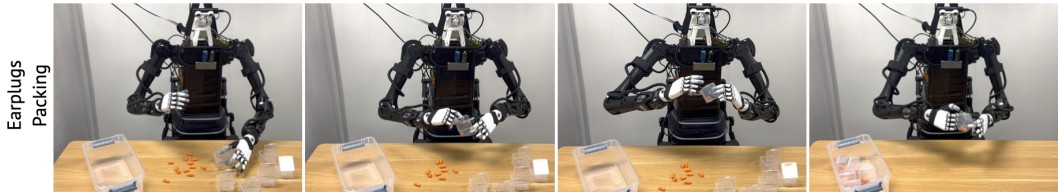

(b) The robot picks randomly placed earplugs on the table and places them into randomly placed latch boxes. The robot needs dexterous bimanual in-hand manipulation and adjustments to properly close the latch box.

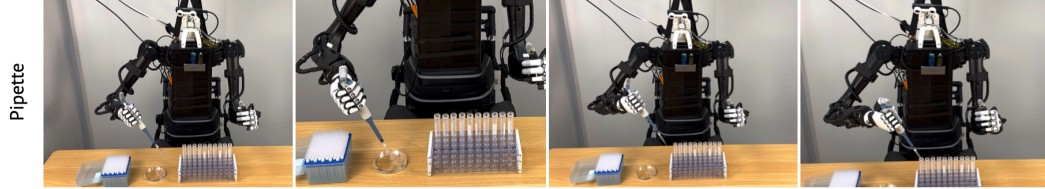

(c) The robot utilizes its thumb DoF to control a pipette to transfer liquid from a petri dish to a centrifuge tube. The diameter of the tube is only $1.5cm$ so it requires high precision to complete the task.

Figure 12: **More teleoperation experiments.** These experiments aim to show our system's reliability and precision for a wide variety of tasks.

| Teleop System | Actuation | Hand | Bimanual | Perception | Remote | Depth |
|---|---|---|---|---|---|---|
| OPEN TEACH[2] | VR Controller | ✓ | ✓ | Direct View+RGB | ✗ | ✓ |
| HATO[3] | VR Tracking | ✓ | ✓ | Direct View | ✗ | ✓ |
| AnyTeleop[6] | RGB(D) Tracking | ✓ | ✗ | Direct View/RGB | ✓ | ✓ |
| Telekinesis[7] | RGB Tracking | ✓ | ✗ | Direct View/RGB | ✓ | ✓ |
| Transteleop[8] | IMU+Depth | ✓ | ✗ | Direct View | ✗ | ✓ |
| ALOHA[12] | Joint Copy | ✗ | ✓ | Direct View | ✗ | ✓ |
| AirExo[13] | Joint Copy | ✗ | ✓ | Direct View | ✗ | ✓ |
| GELLO[15] | Joint Copy | ✗ | ✓ | Direct View | ✗ | ✓ |
| Mobile ALOHA[16] | Joint Copy | ✗ | ✓ | Direct View | ✗ | ✓ |
| DexCap[17] | SLAM+Mocap | ✓ | ✓ | Direct View | ✗ | ✓ |
| Open-TeleVision | VR Tracking | ✓ | ✓ | Stereo | ✓ | ✓ |

Table 7: **Comparing Open-TeleVision's capabilities with prior teleoperation systems.** A more detailed analysis to the contents in this table is in Appendix. B.

