# OpenReview forum: "Open-TeleVision: Teleoperation with Immersive Active Visual Feedback"
_robot-learning.org/CoRL/2024/Conference — CoRL 2024_

### Official Review · Reviewer_QGk6 · 2024-07-21
**Good contribution but advantage is unclear**

**Originality:** 2
**Technical Quality:** 3
**Clarity Of Presentation:** 4
**Potential Impact:** 3
**Recommendation:** 2
**Confidence:** 4

**Review:**

### Strengths
- The authors have carefully developed a VR teleoperation system that controls all finger joints and works with multiple humanoid robots. This requires a high-quality engineering effort and if open-sourced, can be a useful contribution to the research community. Especially if plugins to simulators are added.
- The 2 DoF articulated head is a novel contribution and _might_ reduce the gap between regular human perception and humanoid robot perception during dextrous manipulation tasks.
- The paper is well written and easy to understand.

### Weaknesses
- The paper is differentiated from other closely related work like [2], [3], and [4] by claiming that the articulated neck enables the teleoperator to ensure the optimal viewpoint (L29-31). But the effect of the articulated head is not validated in the experiments. This raises concerns about whether the proposed Open TeleVision system is adding value compared to existing work.
- The teleoperation system is one of the contributions, but its accuracy is not quantitatively measured.
- The experiments do not rigorously test the in-the-wild or generalization performance of the policies. Even though some of the tasks have random starting states, they are discrete (e.g. insert in one of the 6 slots), so easily covered in the training dataset.
- Are separate policies needed for each task? If imitating humans is the broad inspiration, then humans are able to transfer knowledge across skills. What changes to the algorithm would be needed to train a single policy? A short discussion on this would make the paper stronger.

**Quality Of The Limitations Section:**

2

**Questions For Rebuttal:**

- Check the performance of policies without articulated head for the tasks chosen in the experiments.
- Measure the accuracy of the teleoperation system by measuring the difference in the 3D location of various corresponding points on the human and robot arm.
- Please clarify L114. What does "projected from" mean?

**Robotics Focus:**

4

**Summary Of Paper:**

This paper describes a teleoperation system for dexterous robot arms. The system is used to collect human demonstrations for everyday tasks, and policies are learnt to imitate these demonstrations.

**Summary Of Recommendation:**

The paper needs important additional experiments to show the importance of the proposed teleoperation system.

---

### Official Review · Reviewer_xknk · 2024-07-21
**Open-TeleVision: Teleoperation with Immersive Active Visual Feedback**

**Originality:** 3
**Technical Quality:** 4
**Clarity Of Presentation:** 4
**Potential Impact:** 3
**Recommendation:** 3
**Confidence:** 4

**Review:**

The paper presents Open-TeleVision, an immersive teleoperation system that enhances robot control using active visual feedback and motion mirroring with VR devices. This system captures the movements of human operators and transfers them to humanoid robots equipped with multi-finger hands and active cameras. The system's effectiveness is validated through data collection for imitation learning on tasks such as can sorting, insertion, folding, and unloading. The results demonstrate improvements in task performance and generalization capabilities. The system was not accessible during the review period, but the authors have stated that the entire system will be open-source.

The paper is well-structured and provides a comprehensive overview of the system, including hardware setup, software architecture, and experimental validation. The detailed experimental results provide clear evidence of the system's effectiveness in improving teleoperation and imitation learning performance. Stereoscopic vision and active visual feedback are well-motivated and show improvements over baseline methods. The paper would benefit from a more thorough discussion of its limitations and the conditions under which the experiments were conducted.

**Quality Of The Limitations Section:**

2

**Questions For Rebuttal:**

Please make a note of the following:

- The paper would benefit from a more thorough discussion of its limitations and the conditions under which the experiments were conducted.
- Can you provide more details about the potential challenges and solutions for scaling the system to handle more complex and varied tasks?
- The paper mentions that there are significant visual occlusions when using the GR-1 robot with grippers, particularly when the gripper grasps objects, which complicates visual inference. It would be helpful to include a discussion of possible methods to mitigate these challenges.

**Robotics Focus:**

4

**Summary Of Paper:**

The paper introduces Open-TeleVision, which is an immersive teleoperation system for robots utilizing VR devices to mimic human movements and offer stereoscopic visual feedback. It showcases efficient data collection for imitation learning and illustrates enhanced performance on diverse tasks in comparison to alternative methods.

**Summary Of Recommendation:**

The paper introduces an efficient teleoperation system with a significant potential impact. Although the work is of high technical quality and well presented, further revisions are required to improve the discussion of results and limitations. Overall, it makes a valuable contribution to the field.

---

### Official Review · Reviewer_mH7J · 2024-07-21
**Trendy work but missing the tribute to the established field of telerobotics**

**Originality:** 1
**Technical Quality:** 2
**Clarity Of Presentation:** 3
**Potential Impact:** 2
**Recommendation:** 3
**Confidence:** 4

**Review:**

Strengths

- The paper is generally readable in terms of grammar and spellings. The figures are also well drawn.
- The work is quite trendy; it tries to combine imitation learning and teleoperation for bi-manual manipulation.

Weakness

- (major) Novelty of the paper is not clear.

The paper claims the main innovation is in its active vision with VR systems. Unfortunately, when compared to papers [1-4], I could not see the main novelty. Many of these works include active vision with VR, by either mapping the camera movements to the head mounted display or creating VR that can change its viewpoints. I wanted to also point out that related work is not satisfying: (a) one should discuss them rather than listing them, e.g., real robots is studied in [71-84] in order to respect the fact that this current paper builds upon them, and (b) the related work should also discuss traditional teleoperation systems where VR and active vision were always part of. The current discussions seem rather shallow.

- (major) As a system paper, the design motivations seem largely missing, especially in section 2.

I think an important question is "why" certain design choices have been made. E.g., we only consider their active sensing neck,.. then why? The computation of forward kinematics is conducted with pinocchio,.. why? In my opinion, this section should be written as to justify each steps rather than merely listing them. In this way, the paper can serve as a system contribution that passes important lessons to future readers.

- (major) The paper claims open-source framework (with its name open-television) but is not available at the time of the review.

For opensource contributions, the reviewers would like to see the quality of the code, and also costs of the hardware platforms and their expositions. The paper only makes a claim that it will be open-sourced and therefore, it is difficult to credit and judge the potential impact.

- Here are also list of minor points:

1. 2 humanoids,... two 7DoF arms, etc. One could unify this, e.g., two humanoids, ... two seven DoF arms, etc throughout the paper.
2. Tables can have bold marks as to highlight the important findings.
3. For user studies, four participants seems rather weak. User studies are serious matters with standardization within the community. The paper can follow that, and also present qualitative findings.
4. The videos should also present the teleoperation system and highlight, rather than only the robot executing the tasks.

[1] Robust Immersive Telepresence and Mobile Telemanipulation: NimbRo wins ANA Avatar XPRIZE Finals
[2] Virtual Reality via Object Pose Estimation and Active Learning: Realizing Telepresence Robots with Aerial Manipulation Capabilities
[3] Human-Scale Bimanual Haptic Interface
[4] An ecosystem for heterogeneous robotic assistants in caregiving: Core functionalities and use cases

**Quality Of The Limitations Section:**

3

**Questions For Rebuttal:**

In my understanding of the field, telerobotics have been one of the traditional topics, and teaching the robot through teleoperation data have been quite widely studied in early 2010s and also before that. Also, if we look at the DLR systems and also the recent AVATAR challenge, the teleoperation systems have reached impressive performance, which could also directly used for learning. I see ALOHA as an addition because that system is specialized in low cost hardware. In this regard, I wonder what is the contribution of this paper given the developments over last few decades? What makes the presented system specialized for modern imitation learning toolboxes?

**Robotics Focus:**

4

**Summary Of Paper:**

In RSS 2023, a teleoperation system called ALOHA was presented in order to perform end-to-end imitation learning from teleopeartion for bnimanual manipulation. In the current submission, the authors follow this trend and present a system called Open-TeleVision, where the focus is on introducing stereoscopic vision. Several bi-manual manipulation tasks are demonstrated on a real system.

**Summary Of Recommendation:**

For me, the cons outweigh the pros, which I have outlined in the section above. The paper can improve by (a) stating the contribution and system level novelty in more clear way by paying tribute to the established field of telerobotics, (b) providing design decisions upon description of the system and (c) providing the promised open-source work during the review phase. --------------------------------------------------------------------- ( #post rebuttal comment: I have read the revised manuscript and the author rebuttal. I decided to increase my recommendation to weak accept, given that the novelty is more clear, e.g., the argumentation about the costs in the related work seems convincing and the provided system can serve the community. However, the design choices should be more thorough. I get that the paper can be a useful infrastructure to the community, but methodological lessons could be stronger)

---

### Author Rebuttal · Authors · 2024-08-13

### Rebuttal Zip File

- Revised paper with modified part highlighted with red.
- A rebuttal video with additional experiments.

Please also see the comments to individual reviewers for details.

---

### Decision · Program_Chairs · 2024-09-04

**Decision:**

Accept

**Comment:**

**Pre-rebuttal**

This paper receives a mixed review from three reviewers.

On the positive side, reviewer **QGk6** recognizes the outstanding engineering effort put into the system and its potential contribution to the research community if open-sourced. Reviewer **xknk** acknowledges the empirical effectiveness of the system in improving teleoperation and imitation learning performance.

On the negative side, both reviewer **QGk6** and **mH7J** question about the claimed contribution on the active vision piece. Reviewer **QGk6** questions about the validation of head articulation in teleoperation and policy performance, as well as more quantitative evaluation of the teleoperation interface. Reviewer **mH7J** raises concerns about novelty, particularly on the active vision side and also the differentiation to the established telerobotics field. A lack of discussion on certain design choices is also brought up. Reviewer **xknk** mentions the lack of discussion on the limitation side.

---
**Post-rebuttal**

This paper initially received two weak reject and one weak accept. One reviewer acknowledged the merit of the rebuttal and raised the assessment from weak reject to weak accept.

After reading the paper and reviews, AC agrees with the main criticism, including unclear contribution, particularly with respects to the telerobotics field, as well as the missing empirical validation of the active vision component. However, onboard with the reviewer who overturned the rating, AC agrees that the rebuttal has sufficiently addressed the concerns.

Considering the recent research landscape and the growing interest in the teleoperation of bimanual/humanoid robots, AC thereby recommends to accept the paper with the belief that the paper will be of sufficient interest to the CoRL audience.

AC also encourages the authors for a proper open source endeavor if the paper is accepted, since the contribution of the work is largely contingent upon the open source of the proposed system.

Typos:
- [Line 79] "we" -> "We"